# Rhizobiome Transplantation: A Novel Strategy beyond Single-Strain/Consortium Inoculation for Crop Improvement

**DOI:** 10.3390/plants12183226

**Published:** 2023-09-11

**Authors:** Ma. del Carmen Orozco-Mosqueda, Ajay Kumar, Olubukola Oluranti Babalola, Gustavo Santoyo

**Affiliations:** 1Departamento de Ingeniería Bioquímica y Ambiental, Tecnológico Nacional de México en Celaya, Celaya 38010, Guanajuato, Mexico; carmen.orozco@itcelaya.edu.mx; 2Amity Institute of Biotechnology, Amity University, Noida 201303, India; ajaykumar_bhu@yahoo.com; 3Food Security and Safety Focus Area, Faculty of Natural and Agricultural Sciences, North-West University, Private Mail Bag X2046, Mmabatho 2735, South Africa; olubukola.babalola@nwu.ac.za; 4Instituto de Investigaciones Químico-Biológicas, Universidad Michoacana de San Nicolás de Hidalgo, Morelia 58030, Michoacan, Mexico

**Keywords:** rhizobiome, PGPR, biocontrol, plant diseases

## Abstract

The growing human population has a greater demand for food; however, the care and preservation of nature as well as its resources must be considered when fulfilling this demand. An alternative employed in recent decades is the use and application of microbial inoculants, either individually or in consortium. The transplantation of rhizospheric microbiomes (rhizobiome) recently emerged as an additional proposal to protect crops from pathogens. In this review, rhizobiome transplantation was analyzed as an ecological alternative for increasing plant protection and crop production. The differences between single-strain/species inoculation and dual or consortium application were compared. Furthermore, the feasibility of the transplantation of other associated micro-communities, including phyllosphere and endosphere microbiomes, were evaluated. The current and future challenges surrounding rhizobiome transplantation were additionally discussed. In conclusion, rhizobiome transplantation emerges as an attractive alternative that goes beyond single/group inoculation of microbial agents; however, there is still a long way ahead before it can be applied in large-scale agriculture.

## 1. Introduction

A microbiome is defined as a set of microbial communities (and their gene repertoires and interactions) associated with an organism [1]. The use of microbiomes has been proposed for innovating diverse research areas such as bioremediation, primary food production, food storage, waste valorization, wastewater treatment, biofuel production, and medical applications [2]. The human gut microbiome has been recognized for its beneficial roles in metabolism as well as its influence in antagonizing potential pathogens, behavior, and mental health [3,4]. The concept of a microbiome includes all its genomic potential and functions as well as the interactions that exist within its microbial population [5]. As part of this microbiota, they can include species that do not have a beneficial function in the host, that is, those with neutral activity. However, the synergistic relationships that exist in a microbiome and their potential effect on the host can further highlight the importance of ecological theory. This includes concepts such as the holobiont, which emphasizes an ecological unit that includes a host, its associated microbiome, and the ecological interactions of which it is comprised [6,7].

In the case of plant microbiomes, the communities serve functions similar to those of the widely studied human microbiomes [8]. Much of the plant microbiome consists of plant growth-promoting microorganisms (PGPMs), which primarily include bacteria (e.g., bacilli, pseudomonads, rhizobia, etc.) and fungi (e.g., *Trichoderma* spp., arbuscular mycorrhizal fungi, etc.) that are widely recognized for their beneficial effects and applications in sustainable agriculture [9,10]. The stimulation of plant growth can be realized through two mechanisms: direct and indirect. The first includes the action of producing phytohormones or improving the uptake of nutrients by the plant, whereas indirect promotion includes the antagonism and biocontrol of possible plant pathogens, where pathogen inhibition indirectly improves the health and growth of plants [11,12,13].

There is a wide variety of products that offer microbial agents for improving agricultural production in the bioinoculant market [14]. Some are based on a single microbial species to improve the growth and production capabilities of a crop (which can include different plant crops) or antagonize a multitude of pathogens that can inhabit agricultural soil [15,16,17]. However, previous studies have shown that a single inoculation of beneficial bacteria or fungi may be limited under certain environmental conditions, and their colonization capacity may not be as effective, limiting their promoting and protective effects. To overcome this limitation, the application of what is known as a “microbial consortium”, which may include more than one phylogenetic group, has been used. It is important to mention that a microbial consortium usually consists of two or more microorganisms with an active metabolism and potential synergistic functions [18]. This is the case of bacteria–bacteria, bacteria–fungus, fungus–fungus consortia, and other combinations of microorganisms and abiotic amendments [19,20,21].

An advancement in comparison to microbial consortia (which may include two or more species of microorganisms) is the transplantation of an entire microbiome, such as a rhizospheric microbiome (rhizobiome) [22,23]. The rhizobiome may include hundreds or thousands (millions of cells) of beneficial microorganisms from different species associated with a plant host. To highlight their importance, it has been proposed that microorganisms such as mycorrhizal fungi are responsible for providing up to eighty percent of the phosphorus and nitrogen to their plant hosts [24]. Similarly, soil can host a myriad of microorganisms with the potential to be transplanted, as rhizospheric soil can have greater microbial diversity due to the action of the nutrients exuded by the roots of the plant [25]. Therefore, this study reviews a new concept of rhizobiome transplantation, which should contain functional and active microbiota to protect and promote plant growth. The plant microbiota has been defined as the total microbial community’s residents in an ecosystem or associated with a plant host (or any other organism), while the term microbiome also includes the genomic pool and the interactions that can be generated from that microbiota [1,26]. As a plant microbiome engineering strategy, the transplantation of a rhizobiome faces new challenges because of the possibility of its widespread application in agriculture.

## 2. Rhizospheric Soil: A Rich Source of PGPMs

Soil is home to millions of microbial cells and microfauna that perform various ecological functions [27]. However, the soil microbiota utilizes nutrients that are exudated from plant root systems, and these may include volatile and low-weight molecules such as carbohydrates, phenolics, organic acids, and amino acids, among other secondary metabolites, and high-molecular-weight compounds such as proteins and polysaccharides, etc. [28,29]. The area of the soil in which these root exudates can reach and influence the resident microbiota is known as the rhizosphere. Therefore, the size of the rhizosphere depends on the size and architecture of the root system [30].

Interestingly, some rhizosphere molecules can fulfill signaling functions and regulate microbial activity and abundance, such as fumaric acid, which is produced by banana plants to attract specific strains of *Bacillus subtilis* [31]. Other volatile molecules, such as nitric oxide, which is a regulator of plant metabolism, are also key in signaling legume–rhizobia symbiosis [32]. Likewise, molecules such as stringolactones have been recognized to have a key role in the first steps of plant root arbuscular mycorrhizal fungi (AMF) colonization, and they can also act as positive regulators of nodulation by rhizobial species in legumes [33]. These are examples of how root exudates can act as attractants of soil microbiota and how they play an important role in shaping the plant-associated microbiome. In addition, there are various factors that allow the plant to recruit certain microbiota and, in some cases, select those that can improve the performance of the plants to tolerate stressful conditions, such as drought, flood, heavy metal contamination, or salinity [34]. Some works mentioning beneficial bacteria that are common inhabitants of the soil are mentioned in Table 1.

## 3. Single versus Group Inoculation for Crop Improvement

Usually, from the microbial diversity that inhabits the rhizosphere, organisms that act as biocontrol pathogens and/or plant growth promoters are selected. Some of the most beneficial species include the genera *Bacillus* and *Pseudomonas*, which are ubiquitous and highly abundant in extreme soils and environments [39,69]. Other plant-associated genera include *Pantoea* [70], *Burkholderia* [71], *Arthrobacter* [72], *Rouxiella* [66,73], *Serratia* [74], *Streptomycetes* [75], and nitrogen-fixing bacteria known as rhizobia. Fungi, such as Trichoderma, which includes species such as *T. atroviride*, *T. harzianum*, *T. asperellum*, *T. longibrachiatum*, *T. virens*, and *T. viride* [76], have been widely recognized as allies for the control of pathogenic fungi of plants. Other fungal species, such as AMF, although not known for their antimicrobial activities, can improve nutrient uptake by plants and overall soil health. In fact, even though bacteria contain diverse metabolic arsenals to combat plant pathogens directly, they lack the ability of AMF to capture water in degraded soils, therefore improving soil aggregates [77,78].

Several studies have documented that a single inoculation with beneficial bacteria or fungi can improve the health, growth, and production of vegetable crops [79,80,81,82]. Each species can present genetic determinants in its genome that involve different direct plant promotion mechanisms, such as the solubilization of phosphates and volatile organic compounds, production of siderophores (iron chelating agents), and synthesis of phytohormones (e.g., auxins, goberellins, and cytokinins) and ACC deaminase activity. It also has the potential to influence an indirect action of promoting plant growth and health, which may include the production of lytic enzymes, antibiotics, lipopeptides, and other antimicrobial metabolites [83,84,85,86]. However, it is difficult to identify and select a strain with direct and indirect mechanisms for promoting plant growth as well as one that is a good colonizer and competitor in harsh environments such as the rhizosphere [87]. Similarly, some strains can survive in a rhizosphere environment but do not establish closer relationships with the plant, such as colonization of internal or surface tissues. There are no known examples of bacteria or fungi that can colonize different environments; they are usually rhizospheric, phyllospheric, and/or endophytic bacteria, although there are examples of rhizospheric and endophytic colonization [88,89,90]. The greater the flexibility of a microorganism to associate with a plant, the greater its ability to interact closely and exert its PGP traits.

Therefore, it has been proposed that in a microbial consortium, such as bacteria–bacteria [91], bacteria–fungus [92,93], or fungus–fungus [94], a mechanism or lack of activity can be supplied by its action partner when co-inoculated. For example, it has been documented that the bacterium *Bacillus amyloliquefaciens* co-inoculated with arbuscular mycorrhizal fungi (AMF) such as *Rhizophagus* (formerly known as *Glomus*) *intraradices* can protect maize plants against the pathogen *Spodoptera frugiperda* [95]. Similarly, it has been reported that beneficial microorganisms (PGPR and AMF) improve tolerance to water stress in tomato plants, increasing important parameters such as plant growth and the accumulation of osmolytes and minerals and decreasing antioxidant enzymatic activity.

Other comparative studies have also shown that consortium inoculation may be better than a single inoculation with PGPRs. Rice plants (*Oryza sativa* L.), one of the most important crops for the human population worldwide [96], were inoculated with a group of plant growth-promoting *Bacillus strains* ((*Bacillus licheniformis* (A21), *B. haynesii* (EN43), *B. paralicheniformis* (EN107), *B. licheniformis* (EN108), *B. paralicheniformis* (EN121), and *B. haynesii* (EN124)) and showed a higher biomass accumulation and increased grain yield [97]. Similarly, the synergistic action of two plant growth-stimulating bacteria, *Pseudomonas putida* and *Bacillus amyloliquefaciens*, was observed by the increasing tolerance to drought stress in *Cicer arietinum* L. Importantly, growth parameters such as root and shoot length and dry weights of roots and shoots were significantly higher in plants inoculated with the consortium than in the individual PGPR interaction [98]. To read more about this subject, we recommend reading a review recently published by our group [21].

## 4. Plant Microbiome Engineering Strategies

Plant microbiome engineering is defined as the manipulation or modification of plant-associated microbiomes from different interaction zones, mainly in the rhizosphere, phyllosphere, and endosphere [99,100]. In this review, three different strategies were proposed (excluding genetic engineering): single-strain/species inoculation, dual or consortium application, and rhizobiome transplantation. Table 2 illustrates the advantages and disadvantages of each strategy when applied to crops.

Recently, it was proposed that plant microbiome engineering may occur through host-mediated and multiple generations for microbiome selection [99,100]. Therefore, any change in the microbiome can be caused by a single inoculation with a strain [79]. In addition, the inoculation of a single strain can have multiple effects not only on the metabolism or physiology of the host but also on the associated microbiome and the rhizosphere, endosphere, or phyllosphere. For example, Ferrarezi et al. [101] recently investigated the effects of inoculation with the PGPR *Bacillus thuringiensis* RZ2MS9 and *Burkholderia ambifaria* RZ2MS16 (both isolated from the Brazilian Amazon) on the bacterial community of the rhizosphere microbiome and leaves of maize grown in the field and observed substantial differences in the biodiversity richness and community structure among the plant ecosystems analyzed. In corn, the rhizosphere microbiome is also modulated by a single inoculation with *Azospirillum argentinense* Az39. The authors reported that three of the most abundant genera associated with *A. argentinense* Az39 inoculation were *Burkholderia*, *Massilia*, and *Sphingobium*. Unfortunately, beneficial interacting networks were not evaluated to potentially detect synergism between the rhizosphere bacteria and the Az39 strain [102].

In an interesting study by De Zutter et al. [103], inoculation of maize plants with phosphate-solubilizing bacteria led to a functional relapse (phosphorus status) marked by plants with a low-phosphorus status. Similarly, a second change in the rhizobiome was reported, including bacteria of the Azospirillaceae and Rhizobiaceae families. The authors concluded that conventional in vitro screening methods for phosphate-solubilizing bacteria do not produce an sufficient representation of their performance in planta. Other factors such as chemical fertilization have also been reported to affect the endobiome of maize plants without precisely observed beneficial changes [104]. This is important in a context where more research is required on the unpredictable changes that can occur in the plant microbiome. In certain cases, this can affect plant performance.

Our working group recently published a review on rhizobiome engineering, highlighting two strategies for manipulating rhizosphere microbial populations. The first is known as “top down” engineering, which involves genetically modifying the plant so that it releases certain exudates that attract certain groups of microorganisms in the rhizosphere. The generation of this “new rhizobiome” is expected to have benefits in its plant hosts, such as growth promotion and plant protection. The second strategy is defined as “bottom up” engineering, which arises from the idea of modifying the endemic rhizobiome in order to subsequently manipulate the metabolism of the plant, exerting beneficial actions on plant crops, including an increase in production [99]. 

## 5. Rhizosphere Microbiome Transplantation

The transplantation of an entire microbiome, such as the rhizosphere, is a great step in the engineering of the plant microbiome. With this strategy, the complete gene pool can be transferred as well as its functions (biocontrol and promotion of plant growth) and the interactions of that ecosystem to another. Additionally, the rhizobiome transplant is technically a challenging procedure. Figure 1 shows a composite of the workflow required to transplant a rhizobiome, based on the pioneering work of Jiang et al. [23].

It has been widely documented that the rhizosphere can harbor a beneficial microbiome that can be the first barrier to prevent potential pathogens from damaging the host that harbors and nurtures them. For example, Jiang et al. [23] successfully transplanted a rhizobiome with disease-suppressing abilities, for example, against bacterial wilt disease caused by the pathogen *Ralstonia solanacearum*, into plant crops in disease-conductive soils. Interestingly, this suppressive effect (a 47% reduction in disease incidence) was transferred to other soils by applying this protective rhizobiome to 12 Solanaceae eggplant varieties. The authors were able to isolate a few bacterial antagonistic groups (operational taxonomic units) responsible for the suppression of wilt disease, including *Nonomuraea* sp., *Actinoplanes* sp. (Actinobacteria), *Lacibacter* sp., *Fluviicola* sp. and uncultured bacterial taxa (within the families Sphingobacteriales and Chitinophagales), and Ignavibacteria-OPB56 (Bacteroidetes), uncultured *Aquicella* sp., and a bacterium within the family Solimonadaceae (Proteobacteria), among other groups that were correlated with the suppressive soils. A few years ago, similar results were reported in disease-suppressive soils, highlighting the group of fluorescent *Pseudomonas* (γ-Proteobacteria) as that responsible for antagonism, which is widely abundant [105].

Also, the rhizosphere microbiome of barley plants (*Hordeum vulgare*) has been transplanted under greenhouse conditions to improve the immune response and control diseases caused by the fungus *Blumeria graminis* f. sp. *hordei* (powdery mildew). The main idea of this work was to correct the low presence of beneficial microorganisms in a microbial-poor soil, but by transplanting a microbiome, the beneficial effects and interactions are stimulated in such soil where the barley plants grow. Thus, through the sequencing of 16S rRNA gene amplicons, the authors verified that the transplantation of a rhizobiome could be a sustainable strategy to improve the health of plants grown under greenhouse conditions, in particular, on soils with poor microdiversity, since the plants were less susceptible to the fungal pathogen analyzed [106].

The transfer of “smaller” microbiomes (or few microbial species compared to a complete rhizobiome) has also been previously documented, such as the case of the work by Choi et al. [107], who evaluated the hypothesis that the rhizosphere microbiota stimulates the resistance of the tomato (*Solanum lycopersicum* L.) cultivar Hawaii 7996 against bacterial wilt disease. This cultivar is well known for its resistance to soil-borne bacterial wilt caused by *Ralstonia solanacearum*; therefore, through a sterilization process of the endemic microbiota responsible for the suppressive effect, it was lost in the cultivar. Likewise, the authors demonstrated that the transplantation of the microbiota can confer resistance to these plants and that in the process of heat-killing the transplanted microbiota, a protective effect against wilt disease was not observed. The authors also proposed to continue the study of potential genes and their functions as mechanisms that have yet to be characterized, which would be a fundamental part of the resistance to wilt disease of the tomato cultivar Hawaii 7996.

## 6. Current and Future Challenges

The transplantation of microbiomes into humans to improve health is a reality, and its benefits are widely documented. Numerous studies have been published on this subject [3,108,109], but the transplantation of microorganisms residing in an ecosystem, such as the rhizosphere, is limited [23,107]. In addition, the transplantation of a rhizobiome implies different challenges compared to the transplantation of microbiomes in humans, therefore expanding its application in agriculture that demands greater production in a sustainable way and requires great effort on the part of the academic community. Below, we list certain challenges (but not limited to) that the transplantation of rhizobiomes (or microbiomes) must face in the near future.

### 6.1. Phyllosphere or Endosphere Microbiome Transplantation?

Rhizobiome transplantation involves the use of rhizospheric soil and its subsequent application to other plant-growth systems [23]. The use of soil has certain advantages, such as its handling, as it can be transported and sold commercially in bags, boxes, or other containers. However, can a transplant be made from other areas that harbor beneficial microbiota, such as the phyllosphere or endosphere? As previously mentioned, phyllospheric organisms can be sprayed and reside in a superficial environment or even penetrate tissues and inhabit endophytes [101,110,111]. Until now, it has been complicated to achieve transplantation of a complete phyllobiome or endobiome, but perhaps isolating, characterizing, and applying a synthetic phyllobiome/endobiome that harbors a functional microbiota may be an alternative [110,112].

### 6.2. Incompatibility between the Rhizobiome and the Plant

Jousset and Lee [113] proposed that a possible challenge in transplanting rhizobiomes is their incompatibility with the hosts. Some studies have reported that this incompatibility can be a challenge to overcome, as certain PGPBs may have better interactions with certain plant species. As mentioned previously, root exudates can play an important role in the communication and attraction of certain taxonomic groups, such as stringolactones that recruit AMFs [114,115], or compounds such as fumaric acid, which is produced by banana plants to attract specific strains of *Bacillus* [116]. Another example is the production of oxalate in lupine plants, which attracts oxalotrophic bacteria of the genus *Burkholderia*, a nutritional trait widely found in this microbial group [117,118]. Therefore, a compatibility study between the rhizobiome and potential cultures can avoid this inconvenience [119].

### 6.3. How to Preserve a Rhizobiome for Future Applications?

The single or joint application of some species of beneficial organisms, mainly bacteria or fungi, would mean the conservation of the original strains in stocks that can be created in microbiology laboratories that require routine equipment [120]. The above processes must preserve and produce new inoculants each time production is required. In addition, bioencapsulation of soil microorganisms is another option to preserve their viability during storage [121]; however, in the case of a rhizospheric microbiome, it seems that preserving soil (or different soils from several samples) in the laboratory would require specific equipment, such as large cold rooms. In some previous works, the conservation of fungal species has occurred in substrates such as soil or silica gel [122]. However, conserving a microbiome with hundreds or thousands of species requires more long-term studies to better understand the survival requirements (e.g., nutritional, temperature, periods of time, etc.). Another problem is the viability of the rhizobiome, which requires constant screening of the main beneficial microorganisms in the rhizobiome. This prevents the loss of efficacy when the rhizobiome is transplanted into agricultural crops. Previous studies have shown that, in suppressive soils, the number of cells (or colony-forming units) is important for maintaining the suppressive effect of the disease [105]. The same effect is relevant when transplanting plant growth-promoting activities from the rhizobiome.

### 6.4. Transplantation of Plant Growth-Stimulating Traits

To date, only disease-suppressing soil rhizobiome transplants have been explored as a study model, whereas the transplantation of plant growth-promoting rhizobiomes (PGPROME) has been neglected. This characteristic of direct stimulation may include the action of microorganisms either by modulating plant hormone levels or facilitating resource acquisition from the soil [40,123]. These results are close to those published by a research group working with PGPBs. These types of studies require several years [23,113], and if they are carried out in the field, a minimum of two seasons are required to determine the consistency of the results due to the heterogeneity of environmental variables. It would be advisable to carry out mesocosm or greenhouse studies to control these variables, including some types of stress (e.g., drought, salinity, and the presence of heavy metals), that evaluate the behavior of the transplanted rhizobiome. See Figure 2 to summarize the aforementioned challenges.

## 7. Conclusions

Research on the transplantation of rhizospheric microbiomes is in its infancy, but studies such as those of [23] have demonstrated its feasibility. In this review, we list some challenges and areas of opportunity to continue with this area of research, which is not only interesting from an ecological point of view but also technically challenging; it is also necessary to continue with new proposals for agro-sustainable production.

## Figures and Tables

**Figure 1 plants-12-03226-f001:**
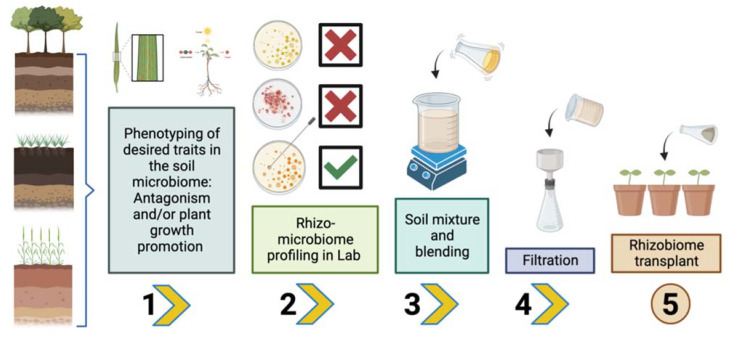
Workflow required to transplant a rhizosphere microbiome (based on Jiang et al. 2022 [23]). In step 1, it is important to discover soils with good microbial diversity and/or that have relevant functions in providing good plant growth or suppressing diseases. Step 2 involves a screening of the microbiota of different soils (preferably several types) with various mechanisms to promote plant growth through direct and indirect action, including the solubilization of elements such as phosphorus, potassium, or zinc and production of phytohormones and production of 1-aminocyclopropane-1-carboxylate (ACC) deaminase or iron-chelating agents such as siderophores, antibiotics, and volatile organic compounds (VOCs), among others. During steps 3 and 4, a mixture of the soil and a filtrate are made to later be transplanted in the crops where the beneficial action of the rhizobiome is to be exerted (step 5).

**Figure 2 plants-12-03226-f002:**
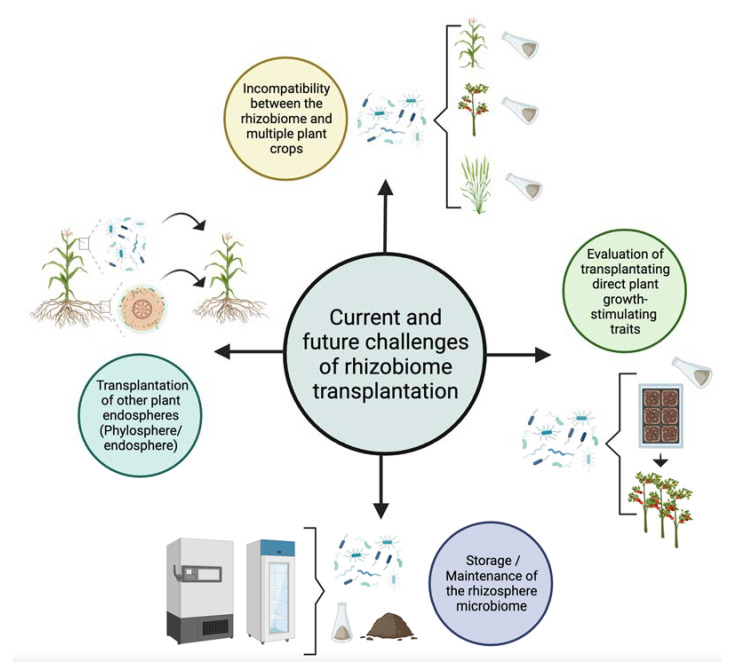
Composite picture of current and future challenges proposed in this work.

**Table 1 plants-12-03226-t001:** Examples of plant growth-promoting and biocontrol rhizobacteria as well as their beneficial activities in different crops.

Bacterial Group/Species	Role	Beneficiated Plant Crop	Reference
*Azospirillum brasilense*, *A. lipoferum*, *Gluconacetobacter diazotrophicus*, *A. brasilense*	Biofertilization and biostimulation	Sugarcane (*Saccharum officinarum*), maize (*Zea mays*), wheat (*Triticum aestivum* L.), rice (*Oryza sativa*)	[35,36,37]
*Bacillus amyloliquefaciens*, *B. aryabhattai*, *B. cereus*, *B. endophyticus*, *B. megaterium*, *B. mojavensis*, *B. cabrialesii*, *B. subtilis*, *B. pumilus*	Biofertilization, bioprotection and biostimulation	Sugarcane (*Saccharum officinarum*), maize (*Zea mays*), wheat (*Triticum aestivum* L.), rice (*Oryza sativa*), stone pine (*Pinus pinea* L.), cucumber (*Cucumis sativus*), chickpea (*Cicer arietinum*), tomato (*Solanum lycopersicum* L.), sweet and chili peppers (*Capsicum annuum* L.), tea plants (*Camellia sinensis*), mung bean (*Vigna radiata*)	[38,39,40,41,42,43]
*Enterobacter oryzae*, *E. asburiae*, *E. ludwigii*, *E. cloacae*, *E. oryziphilus*, *E. oryzendophyticus*	Biofertilization, bioprotection and biostimulation, bioremediation	Mangart and jam (*Acacia acuminate*), wheat (*Triticum aestivum* L.), alfalfa (*Medicago sativa* L.)	[40,44,45,46,47]
*Pantoea agglomerans*, *Pantoea dispersa*, *P. allii*, *P. alaghi*	Biostimulation and bioremediation	Maize (*Zea mays* L.), wheat (*Triticum aestivum* L.)	[48,49,50,51]
*Pseudomonas plecoglossicida*, *P. azotoformans*, *P. fluorescens*, *P. koreensis*, *P. protegens*	Biofertilization, bioprotection and biostimulation, bioremediation	Pearl millet (Pennisetumglaucum), maize (*Zea mays*), wheat (*Triticum aestivum* L.), rice (*Oryza sativa*), tomato (*Solanum lycopersicum* L.), *Medicago truncatula*	[52,53,54,55,56,57]
*Rhizobium meliloti*, *Rhizobium leguminosarum*, *Rhizobium phaseoli*, *R. etli*, *Rhizobium* sp., *Mesorhizobium loti*, *Mesorhizobium cicero*	Biofertilization and biostimulation	Diverse legume plants, soybean (*Glycine max* L.), alfalfa (*Medicago sativa* L.), common bean (*Phaseolus vulgaris*), chickpea (*Cicer arietinum* L.)	[58,59,60,61,62]
*Stenotrophomonas maltophilia*, *S. rhizophila*	Biofertilization and bioprotection	Maize (*Zea mays*), canola (*Brassica napus*), chickpea (*Cicer arietinum* L.), wheat (*Triticum aestivum* L.)	[63,64,65]
*Serratia plymuthica*, *S. marcescens*, *Rouxiella badensis*	Biofertilization, bioremediation, post-harvest bioprotection	Field pumpkin (*Poa pratensis*), summer squash (*Cucurbita pepo*), berries	[66,67,68]

**Table 2 plants-12-03226-t002:** Comparison between plant microbiome engineering strategies and their corresponding advantages and disadvantages.

Strategy	Advantages	Disadvantages
Single-strain or -species inoculation	- Easy manipulation in laboratory; - Practical storage; - Rapid production of bioinoculant; - Reduced costs for production and maintenance; - Practical mixture with other amendments.	- Potential inconsistent results in field; - Lack of some antagonistic and/or plant growth-promoting effects; - Limited beneficial effects on some plant crops. - Easy handling on a large scale
Dual or consortium inoculation	- Easy manipulation in laboratory; - Practical storage; - Reasonable time for production of bioinoculant; - Reasonable costs for production and maintenance.	- Moderate probability of consistent results in field; - Complementation of some antagonistic and/or plant growth-promoting effects between strains; - Beneficial effects on some plant crops; - Might stimulate plant growth under specific harsh conditions.
Rhizobiome transplantation	- High probability of consistent results in field; - Multiple antagonistic and/or plant growth-promoting effects in the rhizobiome; - Application on multiple plant crops; - High potential to stimulate plant growth under adverse environmental conditions.	- Complex manipulation in laboratory; - Complicated storage and handling; - Consumes a lot of time for transplantation; - Elevated costs for production and maintenance. - Difficult large-scale handling

## Data Availability

Not applicable.

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
