# Peer review of "Rhizobiome Transplantation: A Novel Strategy beyond Single-Strain/Consortium Inoculation for Crop Improvement"

_plants, 2023, doi:10.3390/plants12183226_

Round 1

Reviewer 1 Report

This is a very interesting review on the new topic of rhizobiome transplantation. While the scientific community has not yet completely learned how to create the effective microbial consortia, the authors analyze a new or rather alternative way to create effective microbial preparations by rhizobiome transplantation. I have several comments:

1.       It will be helpful to add a definition of the term “microbiota” and describe how it differs from the “microbiome”.

2.       It is also very important to define the term “microbial consortium”. In my opinion, the consortium represents not only several coexisting, but also metabolically integrated species of microorganisms.

3.       Probably, there is a mistake in table 1, rhizobiome transplantation, disadvantages: - High probability of consistent results in field. (consistent instead of inconsistent).

4.       The same for Dual or consortium inoculation.

A little punctuation correction would be helpful.

Author Response

Dear Editors,

Plants

Attached you will find a revised version of our manuscript (Rhizobiome transplantation: a novel strategy beyond single strain/consortium inoculation for crop improvement) with highlighted changes according to the reviewer´s suggestions. We took into considerations ALL changes made by the reviewers, which we appreciate very much. With these new additions, the number of words exceed 4,000, as suggested. The overlap was also rewritten and modified accordingly.  A clean version is also submitted.

We hope that this new version fulfills the reviewers and Editors expectations.

Sincerely,

Gustavo Santoyo

We are answering to reviewers as follows:

Reviewer 1

This is a very interesting review on the new topic of rhizobiome transplantation. While the scientific community has not yet completely learned how to create the effective microbial consortia, the authors analyze a new or rather alternative way to create effective microbial preparations by rhizobiome transplantation. I have several comments:

  1. It will be helpful to add a definition of the term “microbiota” and describe how it differs from the “microbiome”.

RESPONSE: Thank you for your suggestion. It was added the following paragraph with two more references to clarify the term: ¨ The plant microbiota has been defined as the total microbial communities resident in an ecosystem or associated with a plant host (or any other organism), while the microbiome term also includes the genomic pool and the interactions that can be generated from that microbiota [1,23]¨

  1. It is also very important to define the term “microbial consortium”. In my opinion, the consortium represents not only several coexisting, but also metabolically integrated species of microorganisms.

RESPONSE: You are right, The following definition was added: ¨ It is important to mention that a microbial consortium usually consists of two or more microorganisms with an active metabolism and potential synergistic functions [17].¨

  1. Probably, there is a mistake in table 1, rhizobiome transplantation, disadvantages: - High probability of consistent results in field. (consistent instead of inconsistent).

RESPONSE: Thank you very much for your comment, the whole Table was corrected.

  1. The same for Dual or consortium inoculation.

RESPONSE: Thank you very much for your comment, the whole Table was corrected.

Dear reviewer 1, Thank you very much for your comments that for sure, have improved our manuscript.

Reviewer 2 Report

In my opinion, this is more of a mini review than a regular review paper. The authors should make more in-depth analysis of topics before publishing. There are definitely too few concrete examples to support theses included; the background is quite poor (e.g. the title of chaper 2 including words "a rich source of PGPMs" is not supported enought ; I propose to include Table which will contain numerous examples; the same to suport the statement in l. 113). Thus, the whole manuscript needs expansion and better justification.

More detailed comments below:

l.16 In the Abstract section, authors should focus only on the objectives of the review and not discuss them. In relations to this, I propose to omitt „not only plant protection, but also to” and then edit the sentence. E.g. „In this review, rhizobiome transplantation was analyzed as an ecological alternative for stimulation the growth and increasing the crop production. Also, the application of this technic was compared with single strain/species inoculation as well the use of dual or consortium……”.

l.19-20; please omitt „research on” and „rhizobiome” and then edit the sentence

l. 27 – Throughout the manuscript, the authors incorrectly use the word "microbiome" as a synonym for microbiota; please correct it

l. 30 – please, replace wording „and impact” into „as”

l. 32-41; Please do not compare human microbiota function with plant microbiota and develop a thought about the interactions of microbes in the environment that may be related to the topic being described

l. 42 as above

l. 47 biological control of phytopathogens is one of the indirect features that organisms classified as PGP may have; I am giving an example overview on this topic (https://doi.org/10.3390/app12031231), in the following sentences, the authors write about it themselves, so we have some inaccuracy here.

l. 65 needs improvement; in my opinion it should be "An advancement in comparison to microbial consortia"

l. 67-71 is an example of the general remark I made at the beginning

l. 74 maybe replace „owing to” into „because of possibility of”

l. 97 references has been omitted

l. 130 unnecessary repetition of examples of consortia

l. 140-145 this statement does not confirm the introduction made

l. 180 maybe replace „constant” into „a few times repeated” what will be in accordance with Ref.; although the sentence contains true information, such a shortened description of the study may be incomprehensible to the reader; moreover, it seems to me that the entire paragraph does not fit this chapter because it contains the challenges of rhizobiome transplantation and only the next chapter describes this technique (this is another example to support general remarks)

l. 195 In my opinion, the authors should include a description of this technique in the manuscript in addition to Fig. 1; the reader should obtain comprehensive information here and not from Ref.[20].

l. 225 „small microbiome” what does it mean?

l. 237 incorrect number of chapter

l. 240 „transplantation of ecosystem” very unfortunate wording, because an ecosystem is a biocenosis with a biotope

l. 241-244 completely incomprehensible wording

l. 244 style and grammar must be improved; According to „Authors Information” Fig. 2 should be placed after description not before

l. 252 This sentence does not fit to the title of chapter, moreover it is repetition.

l. 258 This thesis should be expanded and some explanation why synthetic systems can be alternative and how to obtain such systems. In its current form the sentence is just speculation with no evidence to back it up.

l. 273 It's entirely a hypothesis; no specific solutions given

l. 277-290 The paragraph does not answer the question in the title

The entire manuscript should be corrected both in terms of cause and effect, as well as language. Each setion should contain an introduction, development and conclusions

Author Response

Dear Editors,

Plants

Attached you will find a revised version of our manuscript (Rhizobiome transplantation: a novel strategy beyond single strain/consortium inoculation for crop improvement) with highlighted changes according to the reviewer´s suggestions. We took into consideration ALL changes made by the reviewers, which we appreciate very much. With these new additions, the number of words exceeds 4,000, as suggested. The overlap was also rewritten and modified accordingly.  A clean version is also submitted.

We hope that this new version fulfills the reviewer's and Editor's expectations.

Sincerely,

Gustavo Santoyo

We are answering to reviewers as follows:

Reviewer 2

In my opinion, this is more of a mini review than a regular review paper. The authors should make more in-depth analysis of topics before publishing. There are definitely too few concrete examples to support theses included; the background is quite poor (e.g. the title of chaper 2 including words "a rich source of PGPMs" is not supported enought ; I propose to include Table which will contain numerous examples; the same to suport the statement in l. 113). Thus, the whole manuscript needs expansion and better justification.

RESPONSE: Thank you very much for your constructive comment, they are taken into full consideration. We agree with you that our work should be a mini-review. And you are right, the topic is relatively new, and very few works are related to the topic of rhizobiome transplantations, but this is what we wanted to express in our work, that is why we see our work more like a perspective review; however, we added several paragraphs to the text, more references, and the Table as you suggested to improve the ms. We hope these additions fulfill your expectations.

More detailed comments below:

l.16 In the Abstract section, authors should focus only on the objectives of the review and not discuss them. In relations to this, I propose to omitt „not only plant protection, but also to” and then edit the sentence. E.g. „In this review, rhizobiome transplantation was analyzed as an ecological alternative for stimulation the growth and increasing the crop production. Also, the application of this technic was compared with single strain/species inoculation as well the use of dual or consortium……”.

RESPONSE: Thank you for your comment. Modified as suggested.

l.19-20; please omitt „research on” and „rhizobiome” and then edit the sentence

RESPONSE: Thank you for your comment. Modified as suggested.

  1. 27 – Throughout the manuscript, the authors incorrectly use the word "microbiome" as a synonym for microbiota; please correct it

RESPONSE: You are right. We added the following paragraphs to clarify this issue and added three more references:

L61-63: ¨ It is important to mention that a microbial consortium usually consists of two or more microorganisms with an active metabolism and potential synergistic functions [17].¨

L74-77: ¨ The plant microbiota has been defined as the total microbial communities resident in an ecosystem or associated with a plant host (or any other organism), while the microbiome term also includes the genomic pool and the interactions that can be generated from that microbiota [1,23]¨

  1. 30 – please, replace wording „and impact” into „as”

RESPONSE: Modified as suggested.

  1. 32-41; Please do not compare human microbiota function with plant microbiota and develop a thought about the interactions of microbes in the environment that may be related to the topic being described

RESPONSE: Thank you for your comment. In fact, we are not the first ones to compare the plant and human microbiota, there are several excellent works that have evaluated this topic, including the concept of ¨holobiont¨ as an ecological unit. This is discussed and references were cited accordingly to justify the introduction.

  1. 42 as above

RESPONSE: Thank you for your comment, but we prefer to leave it as it is.

  1. 47 biological control of phytopathogens is one of the indirect features that organisms classified as PGP may have; I am giving an example overview on this topic (https://doi.org/10.3390/app12031231), in the following sentences, the authors write about it themselves, so we have some inaccuracy here.

RESPONSE: Thank you for your comment. The first sentence was deleted to avoid confusion. We also included the reference suggested!

  1. 65 needs improvement; in my opinion it should be "An advancement in comparison tomicrobial consortia"

RESPONSE: Great suggestion, thanks.

  1. 67-71 is an example of the general remark I made at the beginning

RESPONSE: We added the following paragraph to clarify this issue: To highlight their importance, it has been proposed that fungal microorganisms such as mycorrhizae are responsible for providing up to eighty percent of the phosphorus and nitrogen to their plant hosts [24].¨

  1. 74 maybe replace „owing to” into „because of possibility of”

RESPONSE: Great suggestion, thanks.

  1. 97 references has been omitted

RESPONSE: Reference added. Thanks.

  1. 130 unnecessary repetition of examples of consortia

RESPONSE: We appreciate your comment, however, we consider it is important to highlight the types of inter/intra-kindom consortia.

  1. 140-145 this statement does not confirm the introduction made

RESPONSE: Thank you, A new reference was added to support the statement: Faizal Azizi, M. M., & Lau, H. Y. (2022). Advanced diagnostic approaches developed for the global menace of rice diseases: a review. Canadian Journal of Plant Pathology44(5), 627-651.

  1. 180 maybe replace „constant” into „a few times repeated” what will be in accordance with Ref.; although the sentence contains true information, such a shortened description of the study may be incomprehensible to the reader; moreover, it seems to me that the entire paragraph does not fit this chapter because it contains the challenges of rhizobiome transplantation and only the next chapter describes this technique (this is another example to support general remarks)

RESPONSE: We deleted the word ¨constant¨ to avoid confusion, thanks.

  1. 195 In my opinion, the authors should include a description of this technique in the manuscript in addition to Fig. 1; the reader should obtain comprehensive information here and not from Ref.[20].

RESPONSE: Thank you for your comment. We added the following: ¨ Figure 1. Workflow is required to transplant a rhizosphere microbiome (based on Jiang et al. 2022). In step 1, it is important to discover soils with good microbial diversity and/or that have relevant functions in providing good plant growth or suppressing diseases. In step 2, a screening of the microbiota of different soils (preferably several types) with various mechanisms to promote plant growth through direct and indirect action, including the solubilization of elements such as phosphorus, potassium, or zinc, production of phytohormones, production of 1-aminocyclopropane-1-carboxylate (ACC) deaminase or iron chelating agents such as siderophores, antibiotics, volatile organic compounds (VOCs), among others. During steps 3 and 4, a mixture of the soil and a filtrate are made to later be transplanted in the crops where the beneficial action of the rhizobiome is to be exerted (step 5).  ¨

  1. 225 „small microbiome” what does it mean?

RESPONSE: Thank you, We added the following to clarify it: ¨(or few microbial species compared to a complete rhizobiome)¨

  1. 237 incorrect number of chapter

RESPONSE: Modified as suggested, THANKS!

  1. 240 „transplantation of ecosystem” very unfortunate wording, because an ecosystem is a biocenosis with a biotope

RESPONSE: THANK YOU. We modified the text as follows: ¨ Numerous studies have been published on this subject [3,75,76], but the transplantation of microorganisms residing in an ecosystem, such as the rhizosphere, is limited [23,74].¨

  1. 241-244 completely incomprehensible wording

RESPONSE: The sentence was modified, thanks.

  1. 244 style and grammar must be improved; According to „Authors Information” Fig. 2 should be placed after description not before

RESPONSE: Modified as suggested, thanks.

  1. 252 This sentence does not fit to the title of chapter, moreover it is repetition.

RESPONSE: Perhaps it is repeated, but it is important as an introduction to the subheading and better compare the challenges to transplanting other microbiomes.

  1. 258 This thesis should be expanded and some explanation why synthetic systems can be alternative and how to obtain such systems. In its current form the sentence is just speculation with no evidence to back it up.

RESPONSE: Thank you, We added a couple of references to support our statement.

  1. 273 It's entirely a hypothesis; no specific solutions given

RESPONSE: It is a proposal. We added a reference to support our statement. ¨ Rajeela, T. K., Gopal, M., Gupta, A., Bhat, R., & Thomas, G. V. (2017). Cross-compatibility evaluation of plant growth promoting rhizobacteria of coconut and cocoa on yield and rhizosphere properties of vegetable crops. Biocatalysis and Agricultural Biotechnology9, 67-73.¨

  1. 277-290 The paragraph does not answer the question in the title

RESPONSE: We added the following sentence to further support our idea with one more reference: ¨ In some previous works, the conservation of fungal species has occurred in substrates such as soil or silica gel [89]. However, conserving a microbiome with hundreds or thousands of species requires more long-term studies to better understand the survival requirements (e.g. nutritional, temperature, periods of time, etc.).¨

Dear reviewer 2, Thank you very much for your comments that for sure, have improved our manuscript. We appreciate your time and constructive criticism.

Round 2

Reviewer 1 Report

The paper can be published in present form. 

Author Response

Thank you for all your comments

Reviewer 2 Report

I would like to thank the Authors for considering my comments and comprehensive answers

Minor corrections

Author Response

Thank you for all your comments and suggestions. The new version has included them.